# Intelligent Liver Function Testing (iLFT): An Intelligent Laboratory Approach to Identifying Chronic Liver Disease

**DOI:** 10.3390/diagnostics14090960

**Published:** 2024-05-04

**Authors:** Jennifer Nobes, Damien Leith, Sava Handjiev, John F. Dillon, Ellie Dow

**Affiliations:** 1Department of Blood Sciences, NHS Tayside, Ninewells Hospital, Dundee DD1 9SY, UK; 2Population Health & Genomics, School of Medicine, University of Dundee, Dundee DD1 9SY, UK; 3Department of Gastroenterology and Hepatology, NHS Tayside, Ninewells Hospital, Dundee DD1 9SY, UK; 4Gut Group, School of Medicine, University of Dundee, Dundee DD1 9SY, UK

**Keywords:** automation, algorithm, intelligent, liver function tests, chronic liver disease, diagnosis, management, referral, fibrosis, precision medicine, laboratory

## Abstract

The intelligent Liver Function Testing (iLFT) pathway is a novel, algorithm-based system which provides automated laboratory investigations and clinical feedback on abnormal liver function test (LFT) results from primary care. iLFT was introduced to NHS Tayside, Scotland, in August 2018 in response to vast numbers of abnormal LFTs, many of which were not appropriately investigated, coupled with rising mortality from chronic liver disease. Here, we outline the development and implementation of the iLFT pathway, considering the implications for the diagnostic laboratories, primary care services and specialist hepatology clinics. Additionally, we describe the utility, outcomes and evolution of iLFT, which was used over 11,000 times in its first three years alone. Finally, we will consider the future of iLFT and propose areas where similar ‘intelligent’ approaches could be used to add value to laboratory investigations.

## 1. Introduction

Chronic liver disease (CLD) remains the only major disease area in which the rate of premature mortality (deaths in people under the age of 65 years) has not improved since 1970 [1]. Over 10,000 people die annually from CLD across the United Kingdom, with half of the deaths occurring between 45 and 64 years of age [2], and liver disease is now the second leading cause of years of working life lost in Europe [3]. Globally, over 2 million deaths per year are attributed to CLD (liver cirrhosis, viral hepatitis and liver cancers) [4]. Furthermore, despite the advent of some notable breakthroughs, such as the introduction of highly effective direct-acting antivirals in the treatment of viral hepatitis C [5], current projections are for CLD to become an even greater global health challenge. The prevalence of cirrhosis will continue to increase [6], largely driven by the global increase in metabolic dysfunction-associated steatotic liver disease (MASLD) [7,8], which currently affects over one in three adults and one in ten children globally [9]. Linked to the increasing prevalence of cirrhosis, the incidence of primary liver cancer is also rising rapidly, with projections indicating that there will be around 9700 diagnoses per year by 2040 in the UK. With an average five-year survival of just 13%, this represents a significant national health concern [10]. Globally, the number of new cases and deaths from liver cancer per year is predicted to increase by over 55% by 2040, compared to 2020 [11]. In this context, cost-effective approaches to identify CLD early are essential.

However, challenges exist in identifying CLD. Conventional liver blood tests—commonly referred to by the misnomer ‘liver function tests’ (LFTs) in the UK—are requested ever more commonly [12], but, in isolation, these routine tests lack sufficient sensitivity and specificity to ensure the early diagnosis of liver disease [3,12]. Furthermore, even when abnormal LFTs are identified, they are often inadequately followed up, without further aetiological testing to identify the cause of the abnormality. The Abnormal Liver Function Investigations Evaluation (ALFIE) study, completed in the Tayside region of Scotland, assessed primary care LFT requesting in over 95,000 patients, with a median follow-up of 3.7 years. This study showed that 21.7% of patients had LFT results which included at least one ‘abnormal’ value, and at least one per hundred of the patients included in the study developed liver disease. However, only 50% of those with an abnormal LFTs received any further testing [13]. When abnormal tests are appropriately investigated, only a minority of abnormal results are due to CLD—for example, the BALLETS study, following up patients with at least one abnormal liver test result identified in primary care, demonstrated that less than 5% of people with an abnormal liver test had a specific disease of the liver, and many of those that did were unlikely to require treatment [14]. Therefore, to identify individuals with abnormal LFTs likely to have CLD and determine the cause, additional non-invasive fibrosis testing [15] and comprehensive aetiological screening are required [12]—historically resulting in the need for multiple phlebotomy visits and the manual calculation of fibrosis scores by clinicians. The combination of large numbers of LFTs being requested, abnormal LFTs being under-investigated, and the challenge of distinguishing which abnormal test results are due to significant liver disease, means patients with CLD are often diagnosed at an advanced stage, with more than two thirds of patients hospitalised due to decompensated CLD not previously being referred to a liver clinic [3].

Identifying patients with abnormal LFT results due to potentially serious underlying liver disease and ensuring appropriate further testing and their timely referral to secondary care is therefore highly challenging. One potential solution to this problem is the use of laboratory systems which incorporate artificial intelligence (AI)—the use of software, algorithms or systems which can emulate or mimic human intelligence [16,17]. AI is a broad field, encompassing non-adaptive ‘intelligent’ or ‘expert’ systems (in which humans create decision rules to allow for automated decision-making) through to more complex, adaptive machine learning approaches [16,17]. Use of AI in laboratory medicine is an area of great interest given the wide-ranging benefits it could provide across test requesting, result interpretation, image analysis and prediction modelling [18]. However, there are also significant barriers to the implementation of AI in laboratory medicine, particularly for more complex, adaptive processes such as machine learning. These include inadequate education and training, the need for improved information technology infrastructure (and associated financial investment), concerns regarding data availability and quality (including ethical and data governance issues) and the lack of a robust regulatory framework and standards for laboratory accreditation [17,18,19,20].

Our solution to the challenge of abnormal LFTs is a novel, algorithm-based testing pathway—the intelligent Liver Function Testing (iLFT) pathway—which was developed by laboratory medicine specialists and hepatologists from the National Health Service (NHS) Tayside Health Board and the University of Dundee. In this review, we will provide an overview of the development, evolution and results of iLFT to date, discuss possible future refinements of the platform, and suggest other disease areas in which similar ‘intelligent’ lab approaches may add additional value, including in the future of precision medicine.

The development of iLFT has been described in detail previously [21,22,23], but the key factors will be revisited below.

### 1.1. Development and Validation of the Minimum Diagnostic Criteria

The first major step in the development of iLFT was the creation of ‘minimum diagnostic criteria’ (MDC)—the minimum information required to determine the likely diagnosis and most suitable management pathway. The available data were limited to patient information (age, sex, body mass index (BMI), alcohol intake, presence of metabolic syndrome), blood test results of a so-called ‘liver screen’ for different aetiologies of CLD (listed in Table 1) and calculated fibrosis scores (the fibrosis-4 index (FIB-4) and the non-alcoholic fatty liver disease fibrosis score (NFS)) [23]. The MDC aim to safely stratify patients into three groups: those who have uncomplicated liver disease which can be managed in primary care (for example, alcohol-related liver disease without fibrosis), those in whom the cause of liver dysfunction is unclear but who do not have evidence of a serious liver disease or fibrosis (for example, elevated gamma glutamyl transferase without abnormalities in any aetiology test or fibrosis markers), and those who have significant fibrosis or a cause of liver disease which necessitates specialist hepatology management in secondary care (for example, any patient with advanced fibrosis or specific causes of CLD such as autoimmune hepatitis). Where there is uncertainty, such as indeterminate fibrosis scores, the MDC ‘fail-safe’ and recommend referral for specialist review [23].

These MDC were validated in a population of 323 patients from three Scottish tertiary care hospitals. The diagnosis and referral recommendation suggested by the MDC were compared to expert hepatologist opinions following a review of the case notes. Diagnostic agreement was seen in 82.4% of cases and agreement with the suggested referral recommendation occurred in 91.3% of cases. Reassuringly, only 1.5% of cases were triaged by the criteria to remain in primary care where hepatologist review deemed secondary care assessment necessary [23].

### 1.2. How iLFT Works

If liver dysfunction is suspected, primary care clinicians can request iLFT in the same way they would request standard LFTs, using order communications software. However, when making an iLFT request, they are prompted to enter brief clinical details via ‘tick-boxes’—namely the patient’s recorded body mass index (BMI), maximum weekly alcohol intake in the previous six months (> or ≤14 units) and the presence or absence of features of metabolic syndrome. The order communications system then prompts the requestor to take the required blood samples: two serum separator tubes are required for biochemistry, immunology and virology assays, and a potassium-ethylenediaminetetraacetic acid (K-EDTA) tube is required to obtain the platelet count for indirect fibrosis score calculations. Samples are transported to the laboratory in the usual way.

On arrival at the laboratory the samples are processed and loaded onto the automated track system in the same way as any other blood sample. Initially, the iLFT pathway triggers the analysis of a panel of LFTs, with additional tests cascading automatically to produce a liver disease aetiology screen as abnormalities are detected (see Figure 1 and Table 1). The laboratory software, automation and analytical platforms and methods used in the iLFT pathway are detailed in the Appendix A.

The patient’s demographics and results are combined by the iLFT algorithms, an ‘expert system’, in the laboratory information management system (LIMS) to generate one of thirty-three potential, pre-defined iLFT outcomes, which include a plan detailing the probable diagnosis and any recommended further investigation and management. This is returned to the requestor as a short auto-comment appended to the iLFT test result in the order communications system, and a weblink to a centrally hosted PDF file of the relevant management plan. The plan gives a clear recommendation on the follow-up investigation and whether referral is indicated, based on the probable diagnosis and/or the likely presence and severity of liver fibrosis.

### 1.3. iLFT Reference Ranges

The thresholds used to trigger further investigation are shown in the box in Figure 1. These are defined by our local upper reference limits (URLs), with the notable exception of ALT, for which the iLFT threshold (30 U/L) is lower than our standard local URL of 55 U/L. There is considerable evidence that the URLs for ALT are often above the ‘true healthy value’, likely due to the inclusion of patients with subclinical CLD (such as MASLD) in historical reference populations [24]. In 2002, Prati et al. suggested URLs of 30 and 19 U/L for men and women, respectively [25]. This was recently updated to 42 and 30 U/L based on a new reference population of over 9000 individuals and the International Federation of Clinical Chemistry (IFCC)’s standardized method [26]. It should be noted that the ALT URL used in iLFT is not sex-specific. There is also a wider debate to be had, outside the scope of this review, regarding the lowering of our standard URL for ALT.

Where an elevation of ALP is the only abnormality, i.e., the ALT, GGT and bilirubin fall within the defined limits, a full liver screen will only cascade when the ALP exceeds 200 U/L. For mild elevations between 131 and 200 U/L, an outcome is generated detailing the potential causes (namely, bone sources, drug reactions and MASLD) alongside a recommendation to repeat ALP and GGT testing in three months.

For mild, isolated elevations of total bilirubin (between 22 and 60 μmol/L) a limited cascade occurs, triggering the measurement of direct (conjugated) bilirubin and haptoglobin. Where the results of these confirm a predominantly unconjugated hyperbilirubinemia without evidence of haemolysis, a ‘likely Gilbert syndrome’ outcome is generated. If the total bilirubin exceeds 60 μmol/L, a full liver screen is performed to investigate other potential causes.

### 1.4. Laboratory Requirements

Importantly, iLFT utilises standard laboratory technology in a new way. Electronic test requesting, via order communications software, is key. This allows the transfer of demographic data and clinical information directly into the LIMS via barcode scanning. There is bidirectional transfer of information between the LIMS, middleware software and laboratory automation, allowing additional tests to cascade where any of the initial LFTs are abnormal. In our Blood Sciences laboratory, which has advanced automation linking multiple disciplines, samples move between the various tracked modules and analysers in real time, without manual intervention. The clinical information provided by the requestor, such as body mass index, is automatically resulted against a test code, allowing the LIMS to ‘see’ and manipulate these data. The ‘intelligence’ of iLFT comes from the logic rules written into the LIMS, which culminate in the generation of one or more of the 33 possible iLFT outcomes. iLFT is therefore an example of an ‘expert system’ AI approach, as these logic rules have been constructed and coded by humans to mimic human decision-making. The iLFT outcomes each trigger their own auto-comment and management plan for the requestor.

### 1.5. The iLFT Pilot

The iLFT pilot trial [21] was performed in six local general practices between 2015 and 2016. Three urban and three rural practices participated in this stepped-wedge design study and each contributed patients to the control and intervention arms of the study. To be eligible for inclusion, participants had to be aged 18–75, have no jaundice, no diagnosis of pre-existing liver disease or previous abnormal LFTs, nor have LFTs requested as part of routine drug monitoring. Control patients (*n* = 490) were those with an abnormal LFT (where a result fell outside reference intervals), whereas intervention-arm patients (*n* = 229) were those for whom the iLFT ‘intervention’ was offered to GPs (64 of these 229 iLFTs were abnormal) [21].

The primary outcome was the rate of diagnosis of liver disease in the two arms. Secondary outcomes included the number of patient–primary care contacts from initial LFT to diagnosis, the number of referrals to secondary care for diagnosis and a cost-effectiveness analysis comparing clinical practice at the time with iLFT [21].

The results showed that iLFT assisted primary care General Practitioners [GPs] in making a diagnosis in 67% of cases and increased the rates of liver disease diagnosis by 43%. iLFT increased the rates of patient visits to their GP and GP referrals to secondary care, likely as a result of the increased diagnosis of possible CLD requiring follow up. It was found to be more cost-effective compared to routine clinical practice; the cost per correct diagnosis was GBP 284, with a saving of GBP 3216 per person in the lifetime model. iLFT was received favourably by the GPs who used it during the trial, 21/23 (91.3%) of whom wanted to see it offered as a standard clinical care system [21].

## 2. iLFT in Action

Following the successful pilot, iLFT was fully rolled-out across NHS Tayside primary care in August 2018.

### 2.1. Acceptability, Uptake and Requesting Patterns

Over the first three years (August 2018–July 2021), 11,043 iLFT tests were performed. A total of 8096 (73.3%) cascaded to a full aetiology screen, 707 (6.4%) to a partial aetiology screen and 2240 (20.3%) required no further testing. Following its introduction, requesting rose steadily over the first 18 months before falling at the start of the COVID-19 pandemic (Figure 2). The number of requests subsequently recovered to their pre-COVID-19 levels. The cascade rate was high (70–80%) initially, primarily due to the fact that iLFT was being utilised as a follow-up test in patients with initial abnormal LFTs. However, it is anticipated, as iLFT is used increasingly as a first line test for suspected liver disease, that the cascade rate will decrease.

Regarding acceptability, a second survey was conducted one year after iLFT was made available across NHS Tayside (August 2019). A total of 100 GPs responded, of which 97 had used the iLFT pathway. Of these, 97.9% (*n* = 95) stated they would recommend iLFT to a colleague and found the iLFT outcomes helpful [27].

### 2.2. Outcomes from the First Three Years of iLFT

Between August 2018 and July 2021, 8803 out of 11,043 iLFTs cascaded to further tests, from which a total of 9879 outcomes were generated (a single iLFT can generate more than one outcome). The outcomes generated by iLFT can be broadly grouped into aetiological outcomes, in which a specific likely causative disease process is identified, or descriptive, in which no particular aetiology is identified, and the outcome describes the pattern of abnormality and presence/absence of possible fibrosis. Of the outcomes generated, 4873 (49.3%) were aetiological and 5006 (50.7%) were descriptive. “iL15: Abnormal ALT (<250 U/L) and a negative liver screen without significant fibrosis” was the most frequent outcome—accounting for 2331 outcomes (23.8% of total outcomes). Likely alcohol-related liver disease [ALD], with or without fibrosis (iL04 & iL05), accounted for 1612 (16.5% of outcomes) and likely MASLD, with or without fibrosis (iL16 & iL17), for a further 1390 (14.2% of outcomes). Table 2 details the ten most frequently generated iLFT outcomes, and the frequency of all outcomes is shown in Figure 3.

iLFT identified possible liver fibrosis in 2058 individuals—27.9% of all full cascades and 18.6% of all iLFTs undertaken. This includes 1197 with indeterminate fibrosis scores and 1061 with high-risk scores, suggestive of advanced fibrosis or cirrhosis. iLFT recommended the consideration of referral to secondary care hepatology services in 2837 patients (28.7% of outcomes, 25.7% of iLFTs undertaken), suggesting that 74.3% of patients could safely avoid a hospital referral.

## 3. Evolution of iLFT

Automated pathways such as iLFT have further benefits, including the ability to adapt to the changing demands of the local population and health services. Additionally, the rich dataset they provide is ideal for supporting research and innovation. In this section, we describe how iLFT has evolved to better meet the needs of patients and primary and secondary care services.

### 3.1. Improving Non-Invasive Fibrosis Assessment: The Addition of ELF

The fibrosis assessment in iLFT was initially based on the FIB-4 and NFS scores. These are ‘indirect’ fibrosis markers, comprising simple laboratory measurements and clinical information, which are associated with (though not directly related to) liver fibrogenesis. These utilise low-cost, widely available assays, and have high and low thresholds which have been shown to ‘rule-in’ and ‘rule-out’ fibrosis [28]. However, a significant proportion of patients had values which fell within the indeterminate ‘grey area’ between the rule in/out thresholds. To ensure patient safety, patients with indeterminate fibrosis scores were recommended for referral to hepatology for further fibrosis assessments, increasing the demand on secondary care services.

However, direct fibrosis markers, those which measure substances directly related to the process of fibrogenesis and extracellular matrix (ECM) turnover in the liver, are also available. Whilst many direct fibrosis markers have been described in the literature, few are routinely measured in clinical practice [28,29]. One exception to this is the Enhanced Liver Fibrosis (ELF) score, which has a strong evidence base across multiple aetiologies of CLD [30,31,32,33,34,35]. The ELF score incorporates the measurement of three individual analytes—hyaluronic acid (HA), procollagen III N-terminal peptide (PIIINP) and the tissue inhibitor of matrix metalloproteinase-1 (TIMP-1). The concentrations of the analytes are combined using a formula to give a unitless score which has been validated against the histological staging of fibrosis from liver biopsy. The thresholds provided by the manufacturer (and corroborated independently) equate to absent/mild fibrosis (<7.7), moderate fibrosis (≥7.7 to <9.8), severe fibrosis (≥9.8) and cirrhosis (≥11.3) [36,37].

We performed a pilot study in 2019 to investigate whether ELF testing could be used to help stratify patients with indeterminate fibrosis scores, thus reducing referral rates. Following approval from our local ethics committee, 102 patients with indeterminate FIB-4/NFS were recruited from the iLFT pathway, of which zero samples showed ‘no or mild fibrosis’ (ELF™ <7.7), 47 (46.1%) had ‘moderate fibrosis’ (7.7 ≤ ELF™ < 9.8), and 55 (53.9%) had ‘severe fibrosis’. A Delphi approach was used to decide that ≥9.8 would become the ELF threshold for referral from the iLFT pathway when fibrosis scores were indeterminate, potentially reducing referrals in the indeterminate group by just under half.

In mid-2020, reduced outpatient clinic capacity due to the COVID-19 pandemic, in addition to the increased diagnosis of liver disease from iLFT, caused liver clinic waiting times to increase. For iLFT patients who had been referred for fibrosis assessment due to indeterminate FIB-4 or NFS scores and were still on the waiting list, their ELF was measured utilizing their frozen immunology serum sample (stored for up to two years, as per standard practice). Where ELF was <9.8, patients were removed from the waiting list with an explanatory letter and the option to opt back onto the waiting list if they wished. In total, 602 patients were tested, of whom 40.6% had ELF < 9.8 and were removed from the waiting list. These patients are currently being invited for a follow-up ELF test three years on, to guide their ongoing management.

ELF was fully introduced to the iLFT pathway in July 2020 and automatically reflexes when patients demonstrate indeterminate or high indirect fibrosis scores. We have recently published data demonstrating that the use of ELF as a second-line test for fibrosis in the iLFT pathway reduces referral by one-third. Additionally, very high ELF scores (≥13) provide useful prognostic information—more than one quarter of patients found to have an ELF > 13 will experience an episode of decompensation of their liver disease within 90 days, and over a third will die within the twelve months following the test [38]. We have incorporated these data into the iLFT outcomes—now recommending that requesting clinicians refer these patients urgently for a secondary care hepatology review.

### 3.2. Identification of Patients with Possible Malignancy

A review of the iLFT database highlighted a potential association between deranged LFT, thrombocytosis and the diagnosis of malignancy. On further investigation, it was found that the specific combination of elevated alkaline phosphatase (>130 U/L) and platelets (>400 × 10^9^/L) in individuals over 40 years of age had a positive predictive value (PPV) of 20% for malignancy. When this was tested in a validation cohort of 71,652 patients, the PPV was 30.6%, vastly exceeding the 3% investigation threshold recommended by the National Institute for Health & Care Excellence for malignancy risk and warranting further (routine) investigation [39,40].

This finding has since been incorporated into the iLFT pathway with the creation of a new iLFT outcome, triggered by the finding of a combination of ALP > 130 U/L and platelets > 400 × 10^9^/L, which recommends clinicians to consider malignancy and request appropriate further tests (such as computed tomography imaging of the chest, abdomen and pelvis, or additional tests guided by symptoms or signs, e.g., faecal immunochemical testing or mammography). A subsequent audit has shown that this is successful in widespread use, contributing to the diagnosis of 18 cancers within the first nine months as part of iLFT [data unpublished].

### 3.3. Adoption of New SLD Nomenclature

In July 2023 the global hepatology community announced a paradigm shift in the classification of fatty liver disease. A new umbrella term, steatotic liver disease (SLD), was introduced and non-alcoholic fatty liver disease (NAFLD) was replaced with the new term metabolic dysfunction-associated steatotic liver disease (MASLD) [41]. The previous false dichotomy between metabolic dysfunction and alcohol-associated liver disease, inherent in the NAFLD definition, is addressed in SLD, which includes the categories metabolic dysfunction-associated steatotic liver disease (MASLD), Metabolic and Alcohol-related Liver Disease (MetALD) and alcohol-related liver disease (ALD), representing a continuous spectrum of alcohol intakes. These changes were able to be rapidly incorporated into the iLFT outcomes, ensuring that the advice to requesting clinicians remained up to date. In 2024, this new nomenclature was able to be mapped directly onto existing iLFT outcomes—with MASLD replacing previous NAFLD outcomes, MetALD replacing the previous combined NAFLD and ALD outcome and the ALD outcome remaining the same. Coinciding with the introduction of the updated iLFT outcomes, an educational newsletter was circulated to all iLFT users to explain the rationale for the new nomenclature.

### 3.4. Further Refining the Algorithm

As demonstrated above, we are continuously seeking ways to improve iLFT and planned future work will look to further refine the iLFT algorithm—increasing its diagnostic accuracy and improving its cost-effectiveness.

One area for improvement will be reducing the number of descriptive iLFT outcomes and increasing the proportion and accuracy of aetiological outcomes. As it represents the most frequent iLFT outcome, “iL15: abnormal ALT (<250 U/L) and a negative liver screen without significant fibrosis” is being reviewed as a priority. A local audit determined the majority of these individuals have probable MASLD upon case note review [data unpublished]. In many cases, these iL15 outcomes were not accurately identified as MASLD due to the test requestor failing to indicate that the patient had a history of metabolic syndrome, in cases where such a history was present. Future adaptions to the order communications system, prompting requestors with the full list of all cardiometabolic risk factors for the new MASLD definition at the point of requesting may address this. In addition, a pilot study is proposed to explore adding HbA1c to the iLFT algorithm for those with iL15 outcomes, to identify patients with possible new pre-diabetes or type 2 diabetes diagnoses, which are both significant risk factors for and complications of MASLD [42]. With the recent US Food and Drug Administration (FDA) approval of resmetirom for metabolic dysfunction-associated steatohepatitis (MASH), an inflammatory variant of MASLD with increased progression to cirrhosis, there is even greater importance in diagnosing MASLD [43]. At a health-service level, the improved diagnosis of MASLD could provide additional benefits such as more accurate disease coding and prevalence estimation, helping to inform health policy.

We are also reviewing the clinical utility and cost-effectiveness of the various aetiological tests in the pathway. For example, “iL01: Possible alpha-1 antitrypsin deficiency” occurs frequently, accounting for 5.6% of all iLFT outcomes, triggered by a serum alpha-1-antitrypsin (A1AT) level < 1.0 g/L and initiating reflex A1AT phenotyping. A1AT phenotyping is unavailable in our laboratory and requires us to send away samples to a reference laboratory, incurring significant monetary and time costs to the NHS Tayside laboratory service. However, A1AT-deficiency liver disease is a rare diagnosis [44], and, therefore, the frequency with which it is identified in iLFT suggests a large number of false positive iL01 outcomes are being generated. This may result in unnecessary costs generated from A1AT phenotyping and potentially unnecessary anxiety and confusion for iLFT requestors and patients being given this initial diagnosis. All iL01 outcomes over the first 5 years of iLFT are currently being reviewed—assessing patient outcomes with regard to both liver and lung disease. Potential options to improve the cost-effectiveness and diagnostic accuracy of A1AT within iLFT that are being explored include reducing the cutoff level at which A1AT phenotyping is reflexed or only testing A1AT in patients with elevated fibrosis markers, reducing the number of initial A1AT tests and increasing the pre-test probability of a clinically relevant A1AT phenotype.

More widely, we plan to perform an updated cost analysis of the whole iLFT pathway, which will incorporate pricing changes over recent years related to inflation, the COVID-19 pandemic and the withdrawal of the UK from the European Union. Additionally, this will allow us to formally assess the balance between the addition of ELF, a more expensive laboratory test, against the wider savings to the health service afforded by reducing referrals for secondary care fibrosis assessments.

## 4. Discussion and the Future Direction of Intelligent Laboratory Systems

As described above, the iLFT pathway adds value beyond that achieved by the measurement of standard LFTs and provides an example of what modern, integrated laboratory services can achieve. iLFT extends the role of laboratory testing from generating precise, accurate, quality-assured results alone, to providing practical advice for primary care staff on the interpretation of those results, identification of possible CLD aetiologies and the potential next steps for investigation. Importantly, iLFT can also identify patients in which further investigation or secondary care referral may not be routinely required, reducing unnecessary referrals to secondary care. Crucially, iLFT delivers this in a way which is acceptable and widely adopted by primary care clinicians, reduces the burden of additional blood testing for patients and is cost-effective, saving resources for the NHS. Since its inception, iLFT has proven adaptable—with refinements in the algorithm introduced in response to the COVID-19 pandemic, new disease nomenclature being readily adopted into the existing outputs and new outcomes created where emerging evidence requires it. Finally, the data generated by iLFT itself have allowed new insights such as the prognostic value of ELF testing and the identification of the association between elevated ALP and thrombocytosis and an increased risk of indolent malignancy. iLFT has been widely recognised as an example of good practice and innovation by both hepatology and laboratory medicine organisations. It has already been adopted in other regions of Scotland, and its national roll-out across Scotland is ongoing, with incorporation into the new national LIMS for NHS Scotland laboratories and NHS England adoption being considered.

It should be acknowledged that there are other community pathways for the detection of CLD. A recent narrative systematic review [45] described ten pathways within the UK, of which six were based on abnormal LFTs, two on risk factors, and two on the combination of both risk factors and abnormal LFTs. Whilst some of these pathways also use automation or ‘intelligent laboratory’ practices—such as reflexing ELF where FIB-4 is raised, or the AST:ALT ratio when ALT is elevated—none are as advanced as iLFT, and there are certainly no pathways currently utilising advanced adaptive AI or machine learning techniques. A 2021 survey [46] found that early detection pathways based on the investigation of abnormal LFTs cover only 40% of healthcare regions in the UK, prompting a further campaign by the British Liver Trust to “make early diagnosis of liver disease routine” for all [47]. The national roll-out of iLFT could provide a standardised approach to the investigation of abnormal LFTs, preventing the inequality generated by the current ‘postcode lottery’.

More widely, intelligent approaches to diagnostics—such as iLFT—meet many of the proposals put forward to help support health services such as the NHS in the post-COVID-19 era. One of the seven recommendations from the ‘LSE–Lancet Commission on the future of the NHS’, published in 2021, is to “improve diagnosis, in circumstances where evidence exists to support early diagnosis, for improved outcomes and reduced inequalities” [48]. As discussed earlier, liver disease is a prime example of a significant unmet public health need, in which earlier and improved diagnosis can provide an opportunity for intervention, improving outcomes and saving costs. The LSE-Lancet Commission recognises that increasing diagnostic capability and capacity has an associated cost in terms of equipment and workforce [48]. However, iLFT is an example of a solution which harnesses the technology already available in most standard hospital laboratories and can add value without significant ‘up-front’ financial investments.

Additionally, the LSE-Lancet Commission, along with workforce plans from all four nations of the UK, recognise the role of ‘task-shift’—the transfer of clinical tasks traditionally performed by doctors to other associate and allied health professionals—in sustaining and supporting the health service [48,49,50,51,52,53]. Whilst we will not touch upon the varying opinions regarding task shift here, it must be recognised that blood test requesting and interpretation are increasingly performed by staff other than doctors in primary care within the UK. Solutions such as iLFT provide a standardised approach to a common yet complicated diagnostic problem and provide clear, unambiguous recommendations to the requestor.

The concept of ‘intelligent testing’ is well suited to many disease areas that rely on laboratory results which commonly require reflex or reflective additional testing, and/or interpretive comments. Potential examples for future research include the interpretation of thyroid function tests and the diagnosis and classification of anaemia, with the reflex testing of haematinics in those found to be anaemic and a clinical interpretation of their results, including advice on probable causes and further investigation. Reflexive laboratory testing is also likely to play a key role in ‘precision medicine’—defined by the Council of the European Union as “a medical model using characterization of individuals’ phenotypes and genotypes for tailoring the right therapeutic strategy for the right person at the right time, and/or to determine the predisposition to disease and/or to deliver timely and targeted prevention” [54]. Precision medicine is acknowledged as essential to addressing an aging and multimorbid global population [55]. The iDiabetes platform, which is being trialled in NHS Tayside, is an example of the potential role of intelligent lab testing in precision medicine [56]. This digital precision medicine platform builds on the concept of iLFT, utilising reflexive lab testing to enhance the accuracy of diabetes diagnosis, determine glycaemic control and assess the risk of diabetes end-organ complications (renal disease, heart failure and cardiovascular and steatotic liver disease). The reflexive laboratory results generated will be combined with polygenic risk scores and clinical information automatically imputed from digital clinical systems to provide individualised patient risk and treatment recommendations, accessible by clinicians providing diabetes care but also, crucially, by the patients themselves. iDiabetes will begin testing in a cluster-randomised controlled trial enrolling up to 24,000 people with diabetes in 2024 and, if proven effective, may demonstrate the integral role of intelligent laboratory testing in aiding in the precision medicine of the future.

In conclusion, iLFT is a well-established example of how ‘intelligent’ laboratory medicine can and will be doing more in the 21st century, not only to provide accurate and reliable test results, but aiding directly with the interpretation and diagnosis of those results. Such intelligent platforms should be trialled in and expanded to appropriate new disease areas as a matter of priority—to aid the wider range of healthcare professionals now ordering and interpreting lab results, to increase health care efficiency by reducing unnecessary and duplicate testing and, ideally, to provide results directly to patients, alongside patient-appropriate interpretations. Finally, intelligent lab systems such as iLFT will be integral to providing the comprehensive, patient-centred precision medicine platforms of the future, essential for an increasingly complex, aging and multimorbid global population.

## Figures and Tables

**Figure 1 diagnostics-14-00960-f001:**
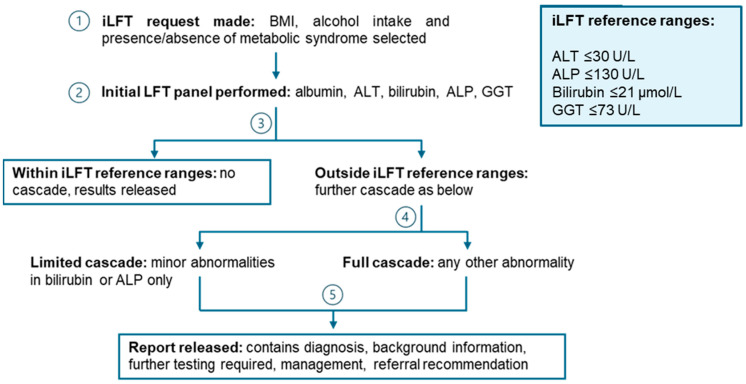
Summary of the iLFT pathway. Abbreviations: BMI, body mass index; ALT, alanine aminotransferase; ALP, alkaline phosphatase; GGT, gamma glutamyltransferase.

**Figure 2 diagnostics-14-00960-f002:**
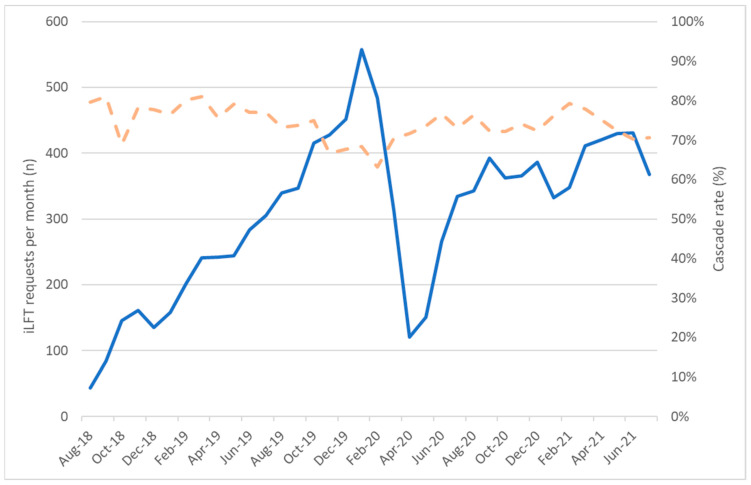
iLFT requesting (solid line) and cascade rates (dashed line) over the first three years of its widespread use.

**Figure 3 diagnostics-14-00960-f003:**
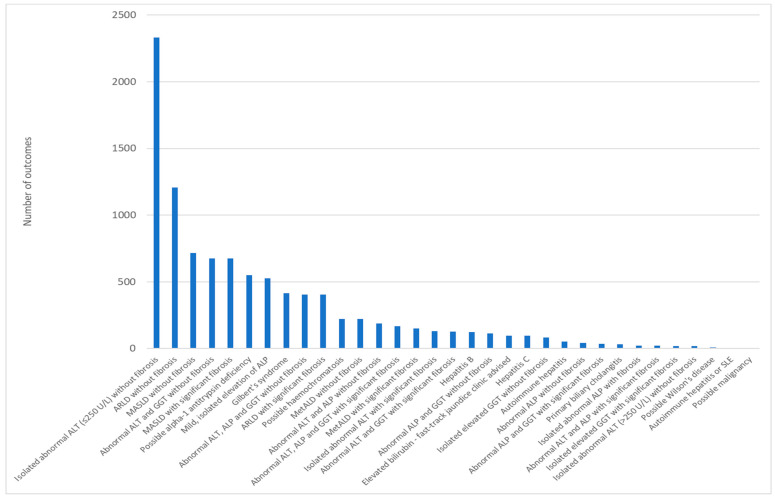
All iLFT outcomes generated in the first three years.

**Table 1 diagnostics-14-00960-t001:** Assays included in the iLFT pathway, along with the clinical justification for each. Abbreviations: HBV, hepatitis B virus; HCV, hepatitis C virus; MASLD, metabolic-associated steatotic liver disease; FIB-4, fibrosis-4 index; NFS, non-alcoholic fatty liver disease score.

**Assay**	**Clinical Justification**
Alanine aminotransferase	Initial assessment for identification of liver dysfunction
Albumin	
Alkaline phosphatase	
Bilirubin	
Gamma-glutamyl transferase	
Haptoglobin	For exclusion of haemolysis in the diagnosis of Gilbert syndrome
Direct (conjugated) bilirubin	
Aspartate aminotransferase	Required for the calculation of fibrosis scores
Iron	Iron studies—for the diagnosis of haemochromatosis
Transferrin	
Percentage saturation of transferrin	
Alpha-1 antitrypsin	For the identification of alpha-1 antitrypsin deficiency
Hepatitis B and C serology	For the identification of HBV/HCV infection (confirmatory tests follow)
Fibrosis-4 index (FIB-4)	Calculated fibrosis score
NAFLD fibrosis score (NFS)	Calculated fibrosis score (used in the algorithms for presumed MASLD outcomes only)
Enhanced liver fibrosis (ELF) score	Direct fibrosis score, reflexes if FIB-4/NFS indeterminate or high
Liver autoantibodies	For the identification of autoimmune hepatitis, systemic lupus erythematosus, or primary biliary cholangitis
**If under 45 years of age:**	
C-reactive protein (CRP)	For the identification of an inflammatory state, if elevated, caeruloplasmin will not be added
Caeruloplasmin	For the diagnosis of Wilson disease

**Table 2 diagnostics-14-00960-t002:** The top ten iLFT outcomes by frequency. Abbreviations: ALT, alanine aminotransferase; ALD, alcohol-related liver disease; MASLD, metabolic-associated steatotic liver disease; GGT, gamma glutamyltransferase; ALP, alkaline phosphatase.

Code	Description	Total (*n*)	Proportion of All Outcomes (%)	Proportion of All iLFT Requests (%)	Liver Clinic Referral Advised?
iL15	Abnormal ALT (<250 U/L) and a negative liver screen without significant fibrosis	2331	23.6	21.1	No
iL05	ALD without significant fibrosis	1208	12.2	10.9	No
iL17	MASLD, simple steatosis without significant fibrosis	715	7.2	6.5	No
iL28	Abnormal ALT and GGT and a negative liver screen without significant fibrosis	676	6.8	6.1	No
iL16	MASLD with significant fibrosis	675	6.8	6.1	Yes
iL01	Possible alpha-1 antitrypsin deficiency	550	5.6	5.0	Dependent on phenotype
iL21	Mild, isolated elevation in ALP	527	5.3	4.8	No
iL06	Likely Gilbert syndrome	413	4.2	3.7	No
iL02	Abnormal ALT, ALP and GGT and a negative liver screen without significant fibrosis	405	4.1	3.7	No
iL04	ALD with significant fibrosis	404	4.1	3.7	Yes

## Data Availability

The datasets presented in this article are not readily available because they are included in ongoing research projects using the iLFT database. However, these will be made available within future full publications. Requests to access the datasets should be directed to Dr Ellie Dow (ellie.dow2@nhs.scot).

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
