# Peer review of "Intelligent Liver Function Testing (iLFT): An Intelligent Laboratory Approach to Identifying Chronic Liver Disease"

_diagnostics, 2024, doi:10.3390/diagnostics14090960_

Round 1

Reviewer 1 Report

Comments and Suggestions for Authors

This is a very well written paper reviewing what has and is a great project. I have but a few minor comments.

1. Line 52. Are the 1 per hundred from the 21.7% with a least 1 abnormal value or overall? My assumption was overall.

2. Paragraph lines 68 - 83 I don't think adds anything in the context of this paper.

3. Line 101 - would be good to flag that 'liver screen' tests are in table

4. line 218 - may not fit with manuscript but interested in the 2 GPs who didn't like it. Any reasons worth noting?

5. line 366 so listing all cardiometabolic markers is planned but not done? Or done but not enough data as yet to determine an effect

Author Response

REVIEWER 1

This is a very well written paper reviewing what has and is a great project. I have but a few minor comments.

  1. Line 52. Are the 1 per hundred from the 21.7% with a least 1 abnormal value or overall? My assumption was overall.

Yes, overall – I have clarified this in the text to prevent confusion.

  1. Paragraph lines 68 - 83 I don't think adds anything in the context of this paper.

I wrote these lines to give context to our idea of iLFT as an ‘intelligent’ system – i.e. of low complexity on the AI continuum, but with good impact despite this and increased suitability for implementation because of this. I have left these sentences in for now based on the comments of the editor and other reviewer, but I am happy to remove them if you feel they are not appropriate.

  1. Line 101 - would be good to flag that 'liver screen' tests are in table

Thank you, this is an excellent suggestion and I have stated “listed in Table 1” here.

  1. line 218 - may not fit with manuscript but interested in the 2 GPs who didn't like it. Any reasons worth noting?

Yes, this is interesting but wasn’t adequately captured by the survey – there were two respondents who selected all the negative outcomes in the survey e.g. “not helpful at all”, “would not recommend to a colleague” and “like nothing about iLFT” but did not elaborate in the free-text sections. We are considering designing a qualitative study to explore the behavioural aspects involved in introducing a new diagnostic pathway and hope to capture and understand the attitudes of these individuals in this work.

  1. line 366 so listing all cardiometabolic markers is planned but not done? Or done but not enough data as yet to determine an effect

Listing all the cardiometabolic features is a potential solution which we are considering but have not yet implemented – I have made this clearer in the text by adding “future”. This seems like a simple solution but adding in additional complexity to the requesting system has not previously been well-received by the requestors. I think it would be useful to consider a trade-off experiment in our qualitative study where we ask participants to weigh-up decisions such as improved accuracy of diagnosis versus increased requesting time and so on to help inform this further. 

Reviewer 2 Report

Comments and Suggestions for Authors

Thank you, I think this is a fantastic and very well written review of iLFT and the potential for clinical and laboratory practice. This sort of process should hopefully revolutionise people's approach to diagnostics and allow us to serve patients better and you describe how you have adapted it so well. Many of the questions I was thinking about you spontaneously answered e.g. why would we test A1AT in all those people as really only a cause of neonatal liver disease isn't it... etc.. 

Only some thoughts, mostly minor or may be not so easy to answer in the review and are discussed elsewhere:

The ranges for iLFT, is is possible to comment on where these came from briefly i.e. local range study, kit insert, etc? These thresholds must affect the outcome/cost effectiveness/clinical effectiveness/referral rates etc so good to have, even if brief, understanding of their origin. this may allow people to understand the work they might need to do to apply this to their own equipment and population. 

Does diagnosing liver disease early reduce patient mortality? Seems a bizarre question and you have addressed hep C for example and you discuss the issue with the 'ALT raised no fibrosis'  comment. Just thought it may be another point to cover, as, with the MASLD renaming, there is one hope that identifying diseases more clearly will allow these patients to access research, be coded better etc. Also some of the liver disease causes harder to 'treat' with tablets, i.e. need lifestyle changes, but may be by identifying people pre-cirrhosis we will better understand how to treat the diseases? Sorry if gone a bit off piste here , just cogitating on the benefit of pointing out the ALT is likely MASLD if that was actually obvious when you go back into and no specific intervention is recommended (besides more funding into public health etc). 

Small changes:

International convention is to remove 's from eponymous syndromes e.g. table 1 Gilbert syndrome, not Gilbert's syndrome. Also Wilson, not Wilson's.

The cost analysis was that inclusive or exclusive of the ELF addition (last time we looked it was an expensive test and does it need another machine or have you got this on your automated platform)?

I know it is appropriate to point readers to previous publications but I wonder if the methods and manufacturers need a word or two (it isn't in 23, 21 is behind a pay wall, but I found it in 22) e.g. whose ALT assay, which ordercomms, which LIMS - particularly when you mention the ranges as hopefully most people will realise they need to know this information. How would you translate this to other manufacturers  and populations - again off piste question, just wondering if a dept was keen but had limited resource how would they go about picking ranges - mentioned above also - but that likely just needs directing people to your previous publications?

Author Response

REVIEWER 2

Thank you, I think this is a fantastic and very well written review of iLFT and the potential for clinical and laboratory practice. This sort of process should hopefully revolutionise people's approach to diagnostics and allow us to serve patients better and you describe how you have adapted it so well. Many of the questions I was thinking about you spontaneously answered e.g. why would we test A1AT in all those people as really only a cause of neonatal liver disease isn't it... etc..

Only some thoughts, mostly minor or may be not so easy to answer in the review and are discussed elsewhere:

The ranges for iLFT, is is possible to comment on where these came from briefly i.e. local range study, kit insert, etc? These thresholds must affect the outcome/cost effectiveness/clinical effectiveness/referral rates etc so good to have, even if brief, understanding of their origin. this may allow people to understand the work they might need to do to apply this to their own equipment and population.

Yes, this is an excellent suggestion. I have added a paragraph after ‘how iLFT works’ discussing the ranges, and, as per your later point, have added supplementary material detailing methods, platforms and software. We are planning to publish data around our choice of ALT upper reference limit in the near future.

Does diagnosing liver disease early reduce patient mortality? Seems a bizarre question and you have addressed hep C for example and you discuss the issue with the 'ALT raised no fibrosis'  comment. Just thought it may be another point to cover, as, with the MASLD renaming, there is one hope that identifying diseases more clearly will allow these patients to access research, be coded better etc. Also some of the liver disease causes harder to 'treat' with tablets, i.e. need lifestyle changes, but may be by identifying people pre-cirrhosis we will better understand how to treat the diseases? Sorry if gone a bit off piste here , just cogitating on the benefit of pointing out the ALT is likely MASLD if that was actually obvious when you go back into and no specific intervention is recommended (besides more funding into public health etc).

This is an important point, and I very much appreciate your cogitation! As you say, there is no benefit per se to identifying liver disease earlier without that translating to an improved clinical outcome via therapeutics, non-pharmacological intervention and/or public health/policy change.  There is obviously a lack of standardisation at the moment which in part is fuelled by the uncertainty regarding what early detection can achieve and the optimal approach to take. This topic could be a review/opinion piece on its own. However, there are benefits as you say with regards to improved MASLD diagnosis – including now the approval of resmetirom for MASH – and I have added a few sentences into the text highlighting these.

Small changes:

International convention is to remove 's from eponymous syndromes e.g. table 1 Gilbert syndrome, not Gilbert's syndrome. Also Wilson, not Wilson's.

Thank you, this has been remedied in the updated manuscript.

The cost analysis was that inclusive or exclusive of the ELF addition (last time we looked it was an expensive test and does it need another machine or have you got this on your automated platform)?

The included cost analysis is from the pilot study and was therefore performed prior to the addition of ELF. However, we are in the planning stages of a further formal cost-effectiveness analysis based on data from the first five years. I have added a paragraph regarding this for clarity. ELF is expensive in terms of laboratory tests (approx. £40/ €45 for us – we already had Siemens analysers so do it in-house) but comparable or cheaper than a Fibroscan appointment.

I know it is appropriate to point readers to previous publications but I wonder if the methods and manufacturers need a word or two (it isn't in 23, 21 is behind a pay wall, but I found it in 22) e.g. whose ALT assay, which ordercomms, which LIMS - particularly when you mention the ranges as hopefully most people will realise they need to know this information. How would you translate this to other manufacturers  and populations - again off piste question, just wondering if a dept was keen but had limited resource how would they go about picking ranges - mentioned above also - but that likely just needs directing people to your previous publications?

Thank you for considering this; as you point out it is included in the JALM paper but I have added information regarding the technical aspects of iLFT as a supplementary material file for interested readers of this article. With regard to other groups choosing their own ranges/thresholds, this is perhaps outside the scope of this review but our previous and upcoming publications may provide useful evidence to inform this.